# Home Immunization with Palivizumab-A Randomized Pilot Study Describing Safety Aspects and Parents’ Preferences

**DOI:** 10.3390/children10020198

**Published:** 2023-01-20

**Authors:** Christina Ebersjö, Eva Berggren Broström, Inger Kull, Anna Lindholm Olinder

**Affiliations:** 1Department of Clinical Science and Education, Södersjukhuset, Karolinska Institute, 171 77 Stockholm, Sweden; 2Sach’s Children and Youth Hospital, Södersjukhuset AB, 118 83 Stockholm, Sweden

**Keywords:** content analysis, home immunization, palivizumab, parents’ preferences, randomized controlled, safety

## Abstract

Among prematurely born infants and newborns with chronic conditions, a respiratory syncytial virus (RSV) infection may cause (re-)admission and later respiratory complications. Therapeutic protection is possible with monthly injections of a specific monoclonal antibody, palivizumab, during RSV season. Standard care is giving up to five injections in clinic-based settings. Immunization at home could be an alternative to standard care for vulnerable infants to reduce the number of revisits and associated risk of RSV infection. The aim of this randomized pilot trial was to evaluate safety aspects and explore parents’ preferences of home versus hospital immunization with palivizumab during one RSV season. Immediate adverse events (AEs) were observed and registered by a pediatric specialist nurse. Late-onset AEs were reported by parents. Parents’ perceptions were collected through a questionnaire and analyzed using content analysis. The study population consisted of 43 infants in 38 families. No immediate AEs occurred. Three late-onset AEs were reported in two infants in the intervention group. Three categories emerged in the content analysis: (1) protect and watch over the infant, (2) optimal health and well-being for the whole family, and (3) avoid suffering for the infant. The study results show that home immunization with palivizumab is feasible if safety aspects are considered and that parental involvement in the choice of place for immunization after a neonatal intensive care experience can be important.

## 1. Introduction

Respiratory syncytial virus (RSV) is the most important cause of acute respiratory infection leading to hospitalization in infants and young children [1]. For premature infants with bronchopulmonary dysplasia (BPD) or newborns with congenital heart disease (CHD), an RSV infection can have severe impact [2]. The risk of rehospitalization increases and infection is associated with acute respiratory complications [3]. Recurrent wheezing and asthma later in childhood have also been found [4]. In addition to the infant’s morbidity, there are several other associated risk factors for developing an RSV infection. These include siblings, crowding [5,6], tobacco exposure [7], and lack of parental knowledge on RSV infection and the benefits of immunization [8]. Risk of severe disease and later complications related to RSV infection are strong reasons to prevent RSV infections in the most vulnerable infants. One method is immunization with palivizumab (Synagis^®^), a monoclonal antibody administered as a monthly intramuscular injection during the winter months [9]. The therapeutic indications for palivizumab are as follows: children born at or before 35 weeks of gestation and less than 6 months of age at the onset of the RSV season; children less than 2 years of age and requiring treatment for BPD within the preceding 6 months; and children less than 2 years of age and with hemodynamically significant CHD. In Sweden, the national guidelines on immunization with palivizumab indicate that this treatment is mainly for extremely preterm infants during their first 2 years of life and newborns with CHD [10]. Standard care involves infants in the immunization program with palivizumab receiving their doses in clinic-based settings. Safety aspects of palivizumab and occurrence of adverse events (AEs) have been described in repeated studies [11,12,13]. Previous studies show that home administration of palivizumab has a positive effect on the possibility to follow the immunization program, a decreased hospitalization rate for RSV, and greater parental satisfaction [14,15,16].

Although the drug palivizumab is well-studied, infants entering the immunization program constitute a fragile group, mainly due to very premature birth. Any additional external stimuli, such as injections, can lead to a need for extra oxygen or apnea, for example. This was important to consider when planning the present study, as it constituted a possible safety risk. 

To our knowledge, there is no information about home administration of palivizumab in the Swedish context, and home administration is not included in the national guidelines. The aim of this randomized pilot trial was to evaluate the safety aspects of home immunization with palivizumab and explore parents’ preferences of home versus hospital treatment.

## 2. Materials and Methods

### 2.1. Design

This pilot study was conducted as a randomized controlled trial among fragile infants randomly assigned to immunization at home or standard care at an outpatient clinic at hospital. To evaluate safety, AEs were registered. To explore parents’ preferences of home and standard care, we used qualitative content analysis of questionnaires. 

### 2.2. Participants

Eligible participants were infants entering the immunization program with palivizumab during one RSV season. Participants were born prematurely or had a CHD. The immunization program followed the national guidelines for immunization with palivizumab, which is indicated for extremely preterm infants born before gestational week 27 during their first 2 years of life and full-term newborns with CHD [10]. 

Participants were randomly assigned to an intervention group (IG) receiving immunization at home or a control group (CG) receiving the immunization at an outpatient clinic or at a hospital (standard care). To avoid randomizing twins into different groups, the participants were grouped in families and sorted in alphabetical order. The first family was randomized to the IG based on a coin toss, the second to the CG, and the rest alternately to the IG or the CG. Infants with previous AEs related to vaccination or immunization were excluded. 

### 2.3. Settings

A specialist nurse with experience in immunization and home care administrated all doses in both groups. Before the first injection, a standardized verbal information—“RSV-infection, effect, benefit and risks of immunization”—was given, in addition to the standard information given at discharge that describes risk factors for transmission of respiratory infections. For calculation of each dose, bodyweight was measured at each visit. Local anesthesia with EMLA^®^ was offered at each immunization. Parents in both groups participated in the same way during administration, holding the child on their knee to be close by and give comfort. At the hospital, another nurse assisted, in accordance with the standard of care, by holding the child’s legs. In homes, parents assisted by keeping their child on their lap in a sitting position and holding the child’s legs. Home visits were booked by phone two weeks in advance to optimize the driving schedule for the nurse. For the CG, each visit was booked approximately four weeks in advance. Visits took place on weekdays. The costs of the drug, palivizumab, was covered by the clinic.

### 2.4. Ethics

The study was conducted in accordance with the Declaration of Helsinki and approved by the Institutional Review Board in Stockholm, Sweden, (2013/286-31/3, Approval date: 22 April 2013). Parents gave written informed consent for inclusion before participating in the study.

### 2.5. Safety Aspects

There was an observation period of 20 min after administration in case of any immediate AE for both groups. For the IG, the nurse stayed in the participant’s home for assessment, with emergency care equipment. If a participant had an ongoing infection, the dose was postponed. In case of a suspected RSV infection, a rapid RSV test (Meridian Bioscience Incorporated, Cincinnati, OH, USA) was used for direct analysis. RSV testing was also conducted in the case of a medical visit or hospitalization due to acute lower respiratory infection. The principal investigator (E.B.B.) had medical responsibility for the study performance and reporting of AEs to the Swedish Medical Product Agency (MPA), in accordance with standard procedure. 

### 2.6. Data Collection

For evaluation of safety aspects, AEs during the observation periods were registered by the study nurse and later AEs were reported by the parents. The characteristics of the participants were collected from hospital records and through a study-specific questionnaire filled out by the parents. To collect parental perceptions of home administration, they were asked to hypothetically choose between immunization at hospital or home and to motivate their choice. A questionnaire was distributed during the last visit, to be answered at home and returned in a prepaid envelope. One reminder was sent after two weeks. An individual code was used in the analysis. 

### 2.7. Data Analysis

An AE was defined as an unexpected event after administration of palivizumab that may or may not have had a causal relationship with the drug. The degree of each AE was assessed as one of three levels: mild, moderate, or severe. 

Qualitative content analysis was used for free-text data from the questionnaires. Content analysis is a research technique useful for exploring people’s experiences of particular phenomena, and its goal is to provide knowledge and understanding of the phenomena being studied [17]. The verbatim-transcribed text was read through several times for the researchers to obtain a sense of the whole (Phase 1), after which meaning units were identified (Phase 2). These meaning units were condensed into descriptions close to the text (Phase 3) and then into interpretations of the underlying meanings (Phase 4). The first, third, and fourth authors abstracted the condensed meaning units into subcategories (Phase 5) and unified them into categories (Phase 6).

### 2.8. Trustworthiness 

Validity in the qualitative analysis was ensured by applying a common analysis process and describing it in detail [17]. Quotations were used to illustrate the results and give the readers an opportunity to evaluate the concordance between the participants’ answers and the categories identified. The pre-understanding of the studied subject comes from two of the authors’ professional occupations as a neonatologist and a registered nurse, respectively, both specialists in pediatric and neonatal care (E.B.B. and C.E.). The other two authors (I.K. and A.L.O.) have less pre-understanding in the area, which might add to the objectivity in this study. All authors were female.

## 3. Results

In total, 46 infants met the inclusion criteria. Of the total families, 3 declined to participate, and 43 infants in 38 families were randomly assigned to the IG receiving immunization at home (23 infants) or the CG with standard care (20 infants). No family withdrew consent. The two groups were similar regarding gestational age, gestational weight, sex, older siblings, and age (Table 1). However, some differences were observed between the groups. In the IG, two participants had ongoing oxygen treatment and three had started kindergarten vs. none in the CG. Eight infants in the IG were in their second season in the immunization program vs. two in the CG. Table 1 shows that the two groups were similar regarding gestational age, gestational weight, sex, older siblings, and age.

### 3.1. Safety Aspects

No AEs occurred during the observation periods in either the IG or the CG. Three AEs were reported by parents after the observation periods in two infants in the IG. One infant who was barely 2 years old had spontaneously recovering urticaria on the upper legs, which appeared the day after the third of five doses. After safety discussions, no changes were made in the administration plan for the final two doses. The infant was born extremely preterm and was in the immunization program for a second season. The second infant had increased oxygen need and fatigue and was subfebrile following the third dose after the observation period and had a similar but much milder reaction following the fourth dose. Both reactions occurred approximately 5 h after the dose. After safety discussions, the fourth and fifth doses were administered at a hospital with an extended observation time (2 h) and a physician present. This infant was nearly 2 months old, with persistent pulmonary hypertension and had recovered from a lower respiratory tract infection shortly before the third dose. The three AEs were assessed as moderate and reported to the MPA. Some mild events were observed, such as local reactions, fatigue, and irritability. None of the participants were infected with RSV during the immunization period. 

### 3.2. Parental Preferences and Motives

All parents in the IG and fourteen of twenty parents in the CG stated that they would prefer immunization at home if they were allowed to choose, but for different reasons. The analyses of the data revealed three core categories with subcategories (Table 2).

#### 3.2.1. Protect and Watch over the Infant

##### The Clinic-Based Settings’ Environment as a Threat and Protection

One underlying concern related to infant safety and health and associated with the clinic-based settings’ environment was identified. The environment was described as a risk because the infant could be exposed to various illnesses and subsequent unnecessary suffering. A visit to the hospital was associated with involuntary contact with people in the waiting room who could infect the infant. Parental concerns regarding infection were explained by a strong belief that their infants were highly susceptible to infection. The environment was described as crowded, without any possibility to avoid the risk of infection: 

“When our child got this vaccination last year, he became ill at every visit to the hospital, which was really difficult. He was ill 2–3 weeks after each injection. Colds + 3 stomach ailments.” (CG, 303); “Since the child is susceptible to infections, you want to avoid being in hospital as best you can.” (IG, 401).

One aspect important to parental preferences regarding location was access to healthcare resources, especially in the case of a severe reaction. For these parents, proximity to professional help was a priority. By choosing immunization at the hospital, these parents felt that they could avoid harm if the infant needed help: “The hospital is safest. They were very kind and helpful.” (CG, 314).

Others based their choice on a desire to receive additional check-ups and medical consultations: “Every time we got Synagis, we met our primary doctor. So, if not for Synagis, we would need to go to hospital anyway, to meet the doctor.” (CG, 302).

##### Avoid Public Places

A parental concern regarding the risk of being infected in situations involving involuntary contact with other people was identified. Public transportation was identified as a risk factor, since it could be difficult to keep distance from other people there. Avoiding crowds and public places had been familiar recommendations since childbirth. Parents who were dependent on public transportation for getting to the hospital thus saw a solution in the possibility to choose immunization at home: “Avoiding public transportation and unnecessary risks of infection.” (IG, 418); “… he is extremely vulnerable, so we avoid going with him to public places where there is a risk of being infected.” (IG, 415).

#### 3.2.2. Optimal Health and Well-Being for the Whole Family

##### Functional Living Conditions

Parental experiences of a long hospital stay were associated with an increased need for a calm home environment with familiar routines. They had struggled to settle into everyday life at home with the infant after discharge: “We’ve already spent so much time at the hospital. It’s nice not having to go there …” (CG, 313).

A visit to the hospital was associated with practical considerations and involved great effort and anxiety: “Our child had many visits to the hospital every week. It felt both good and safe to have this stressful procedure done at home.” (IG, 421).

Regardless of group affiliation, the inevitable travel between home and hospital was described as time-consuming and exhausting: “Because we live out in the country with a long distance to the hospital, and so we can avoid the long waiting time …” (CG, 307).

Parents with sick infants or twins described these journeys as overwhelming. Parents who were home alone with infants in the daytime lacked the support they needed: “Sometimes it’s a huge project to get to the hospital if you have a sick child …” (IG, 416); “We have twins so it’s more practical, easier to be at home when I’m alone with them during the day …” (IG, 405/406); “Since we have twins, it’s so much easier if we don’t have to go to the hospital.” (CG, 305/306).

A need for medical equipment could contribute to complicating the journey back and forth between home and hospital: “… if you’re getting oxygen, it is a huge project to get to the hospital.” (IG, 402).

##### Parental Involvement in Decision-Making

A desire to be involved in decision-making about the infant’s healthcare had an impact on parents’ choice of location for immunization. Parents perceived a lack of communication between themselves and healthcare personnel. Their experience was that it was impossible to coordinate the infant’s immunization with other follow-ups: 

“We are already busy with hospital visits for our children and it has not been possible to coordinate.” (CG, 315/316).

They pointed out that more flexibility in choosing between home and hospital was desirable: “It depends on the circumstances. At hospital if … it can be coordinated with other follow-up visits. At home, if you can get an exact time.” (CG, 304).

Struggling to be more involved was also associated with a wish that individual circumstances could play a greater role in decisions. Parents felt that the decision on the place of immunization should be based on information such as financial circumstances, access to a car, and their own perceptions regarding the infant’s fragility: “For convenience, not everyone has a car and can afford parking. I went by car and could afford the parking fee …” (CG, 312).

Parents also perceived that healthcare personnel should prioritize and determine which infants needed immunization at home on an individual basis: “It’s important that the hospital prioritizes the sickest infants for immunization at home. We have experience from both our children getting Synagis and we ourselves think that our second child was in much greater need of getting the injection at home.” (IG, 416).

#### 3.2.3. Avoid Suffering for the Infant

##### The Home Environment as a Secure and Safe Place

The analysis revealed a perception that the environment was important for the infant’s experience of immunization. The home environment was perceived as a safe and secure place, helping the infant cope with stressful and painful situations: “I believe it can be more secure for the child, especially immediately after the injection when it needs consolation. The child feels safer at home.” (CG,309); “I also believe that my child felt safer at home. A safer child contributes to less pain.” (CG, 313).

The analysis identified parental perceptions that the home environment also had a positive impact for the parents themselves. Being at home in a secure and safe place helped parents be calm, which in turn had a positive impact and calming influence on the infant: “The child and the parents are in a safe environment and calm …” (IG, 418); “Children + parents are more relaxed and the whole ‘vaccination thing’ is experienced as calmer in their home environment.” (IG, 402).

The analysis revealed that parents with experience from the previous season of the immunization program at the hospital perceived a difference in the infant’s behavior during immunization and that the home environment affected the infant positively compared with their experience from the preceding season: “Much more harmonious and calmer. Nice to avoid the hospital environment … it has gone really well for the children to get the injections.” (IG, 411/412).

##### The Infant’s Sensitivity to Disturbances

Parental concerns about the infant’s sensitivity to disturbances in daily care routines also influenced the parents’ hypothetical choice of place for immunization. It was considered difficult to avoid a negative impact on feeding and sleeping routines from a visit to the hospital: “And our child seems calmer at home, and it is easier to plan sleep and food.” (IG, 417).

Many infants had already had long hospital stays, and it was considered important to avoid the hospital as much as possible to promote recovery and development. Parents also considered their infants’ continued need for healthcare to be a reason to minimize revisits as much as possible: “… he is very sensitive, so we avoid going with him to public places …” (IG, 415); “… we wouldn’t have to go to the hospital unnecessarily with a newly operated baby.” (CG, 318).

## 4. Discussion

The aim of this randomized pilot study was to evaluate safety aspects and explore parents’ preferences of home versus hospital immunization with palivizumab. In this pilot study, no immediate AEs appeared after immunization. The study showed that a majority of parents would prefer immunization at home if they were allowed to choose. 

None of the infants in the study suffered any side effects during the observation period of 20 min, but three AEs were observed in two infants in the IG after the observation period. This result is in line with previous studies on the safety and tolerability of palivizumab for RSV prophylaxis in high-risk children [11,12,13,18,19,20]. Adverse immune response to humanized antibodies is a known risk [21], especially in the case of episodic treatment [22], although few incidents are reported for palivizumab [23]. However, serious AEs do occur and must be considered when deciding on the length of the observation period, regardless of if immunization takes place at home or in a clinic-based setting 

The present study shows that parents’ choice between receiving the immunization at home or in a clinic-based setting was influenced by their wish to protect their infant. A protective role may be a natural response after a traumatic childbirth [24]. Some parents in our study were affected by negative experiences from a previous season in the immunization program, when each visit was perceived as an inevitable threat. As expressed by Lasiuk: “when the infant’s health and well-being is tentative, the threat remains omnipresent” [24] (p. 7). However, a protective approach could have lasting negative consequences for both the parent and infant, especially among parents of frail infants. Parental overprotection can be associated with peer victimization during childhood and adolescence, leading to increased risk of anxiety disorder in adulthood [25]. 

The parents’ hypothetical choice of location was affected by their perceived need to be close to medical resources in case of any adverse reactions. Previous reports describe parental uncertainty in taking care of their children [26], apprehension about infant frailty, and a continued need for professional support after discharge [27]. This finding highlights the need to be open to the fact that parents have different needs in planning care. 

A need to avoid public places influenced the preference of location. Public places such as hospitals and public transportation are known to increase the risk of transmission of respiratory infections, especially for vulnerable infants born prematurely, with BPD, or with CHD [5]. Home immunization could be a relief and reduce child-related stress among parents of these high-risk, vulnerable infants [28]. 

Achieving an ability to deal with day-to-day life after a neonatal intensive care experience can take years [29], with long-term impact on the family [30]. Child-related anxiety during an ill infant’s first year of life can also affect a family’s well-being [31] regardless of the severity of the infant’s illness [32]. A continued need for extra oxygenation therapy has been described as being associated with an adverse impact on families’ daily lives [33]. Parenting with responsibility for a medically fragile infant at home in the aftermath of neonatal intensive care focuses attention on the child’s health and vulnerability.

Parents’ needs of communication and involvement in care have been identified already during neonatal intensive care [34,35] and after discharge [36]. Lack of communication can give parents a feeling of exclusion or isolation and raise concerns about the infant’s health status [35]. Improved communication with a focus on the family’s perspective [37] in addition to active involvement are described as essential for individuals with long-term diseases [36]. Financial circumstances may affect a family’s possibility to revisit the hospital when traveling by public transportation is the only option. Difficulties with transportation to follow-up visits have previously been described and are related to infants’ health status [38]. Transportation to the clinic-based settings can require a great effort on the part of parents during the immunization program, which requires multiple revisits. Most of the parents preferred immunization at home. However, a higher cost may be associated with home immunization. The cost effectiveness of RSV prophylaxis has been analyzed from different aspects over the years [28,39,40,41]. This was not included in the purpose of present study. Nevertheless, an unpublished cost estimate was made. Our calculation showed a greater cost when administration was conducted at home, related to a higher assumed waste of the drug. The cost of nursing and the time cost for the family were calculated as less. This is in line with previous reports [28]. 

The familiar home environment had a positive impact on parents and makes the infant calmer during the painful immunization procedure. Previous reports describe the importance of avoiding negative stress linked to painful procedures. The reports state that repeatedly experiencing painful procedures will have a negative impact on an infant’s cognitive and motor development [42,43]. Parental involvement and positive interactions with the child during painful procedures has been associated with fewer visible signs of pain in the child [44,45].

The choice between immunization at home and at the clinic-based setting was influenced by the parents’ perception of the infant’s sensitivity to disturbances. It can be assumed that parents’ perceptions of their infant’s sensitivity arise from their neonatal intensive care experiences. Reports describe that a parental perception of high child vulnerability can negatively affect the infant’s development at an adjusted age of 1 year [45]. Interventions like the one described in the present study might facilitate and be valuable for the infant’s development during the first years of life.

## 5. Strengths and Limitations

The number of available participants during this single season was larger than in several previous seasons, which gives strength to the pilot study results.

All doses for both groups were administrated by the same nurse, which adds strength to the pilot study results.

The study population consisted of families both with and without previous experience of immunization, which adds further strength to the pilot study results.

The control group contained fewer families with experience from previous immunization seasons, which is a limitation of the pilot study.

The study was performed at one (1) hospital, which may be a limitation of the pilot study.

## 6. Conclusions

This pilot study indicates that home immunization with palivizumab may be feasible if consideration is paid to safety aspects and the risk of AE. The pilot study also indicates that parents may prefer immunization at home if they were allowed to choose.

## 7. Implications for Practice

Caring for a vulnerable infant after discharge from neonatal intensive care is a demanding task for parents. It is important for healthcare personnel to reflect on each family’s situation to figure out how to facilitate for the family during their infant’s first years of life. One way might be to offer home immunization with palivizumab.

## Figures and Tables

**Table 1 children-10-00198-t001:** Characteristics of participants.

Characteristics	All *n* = 43	IG *n* = 23	CG *n* = 20
Gestational age, weeks, median (range)	27 (23–41)	26 (23–40)	27.5 (24–41)
Gestational weight, kg, median, (range)	0.935 (0.482–3.65)	0.915 (0.482–3.63)	1.075 (0.656–3.65)
Sex female/male (*n*)	21/22	12/11	9/11
Age at start of immunization, months, uncorrected median (range)	8 (1.9-25)	8 (2.5–20.1)	8 (1.9–25)
BPD diagnosis (*n*)	34	20	14
CHD diagnosis or other circulatory disorder (*n*)	7	2	5
Other diagnosis * (*n*)	2	1	1
Medication for respiratory a/o circulatory disorder (*n*)	16	8	8
Oxygen treatment (*n*)	2	2	0
Second immunization period (*n*)	10	8	2
Verified RSV infection during immunization period (*n*)	0	0	0
Older siblings (yes, *n*)	20	11	9
Exposure to tobacco in home (*n*)	3	1	2
Smoking mother during pregnancy and/or breastfeeding (*n*)	1	0	1
Starting kindergarten during studied season (*n*)	3	3	0

Abbreviations: IG—intervention group; CG—control group; n—number of participants; BPD—bronchopulmonary dysplasia; CHD—congenital heart disease. * One child born premature at gestational age 27 weeks who did not have BPD; one child born premature with multiple diseases, but not BPD or CHD.

**Table 2 children-10-00198-t002:** Subcategories and core categories.

Subcategories	Core Categories
The clinic-based settings’ environment as a threat and protection	Protect and watch over the infant
Avoid public places	
Functional living conditions	Optimal health and well-being for the whole family
Parental involvement in decision-making	
The home environment as a secure and safe place	Avoid suffering for the infant
The infant’s sensitivity to disturbances	

## Data Availability

The data presented in this study are available in Sweden on request from the corresponding author. The data are not publicly available due to the risk of identification.

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
