# Peer review of "Home Immunization with Palivizumab-A Randomized Pilot Study Describing Safety Aspects and Parents’ Preferences"

_children, 2023, doi:10.3390/children10020198_

Round 1

Reviewer 1 Report

This is a well written paper concerning home immunization with palizumab.  Home immunization is not a novel idea and has been practiced extensively for greater than 15 years (as the references demonstrate).  There is additional information concerning parent's perceptions that is of interest to the readers and for that reason I believe this papers merits publication.

I would alter the many sentences that refer to "hospital visits" and change them to "office or clinic-based settings" Standard care is not giving injections at "hospital" visits and this could be misleading to readers. These sentences are lines 14, 50, 184, 349. 

Line 78 an "outpatient clinic at hospital" is a more appropriate description.

Author Response

Thank you and the reviewers for the valuable comments on our manuscript. We are happy that you found the manuscript well written and to be of interest for readers. Please find below our respond point by point. As requested, the revisions made to the manuscript are marked up using the “Track Changes” function.

Reviewer’s comments to authors and our response

Reviewer 1 comments:

This is a well written paper concerning home immunization with palizumab. Home immunization is not a novel idea and has been practiced extensively for greater than 15 years (as the references demonstrate). There is additional information concerning parent's perceptions that is of interest to the readers and for that reason I believe this papers merits publication.

Authors’ reply: Thank you for this positive feedback.

I would alter the many sentences that refer to "hospital visits" and change them to "office or clinic-based settings" Standard care is not giving injections at "hospital" visits and this could be misleading to readers. These sentences are lines 14, 50, 184, 349.  

Authors’ reply: Sentences are changes to clinic-based settings as suggested ( lines 15, 52, 70, 184,187,189,309, 211, 345, 361).

Line 78 an "outpatient clinic at hospital" is a more appropriate description.

Authors’ reply: We have changed to “outpatient clinic at hospital”, according to Your suggestion (line 70/71 and 81/82).

Please let us know if this is enough clarifying.                                                                               Thank you very much for your valuable feedback!

Reviewer 2 Report

This paper reports a pilot study that was randomized. The total number of patients in the trial was 43, which is within the typical range for a pilot study.  On page 2, line 67, the authors mention that it is a pilot study, but that is the only location I found that the "pilot study" aspect of the study was mentioned.   The title of the study gives an impression that it was a randomized controlled trial that provided definitive conclusions.  However, the small sample size of the pilot study does not provide data to derive statistically significant conclusions.     I suggest the following revisions in the paper:   (I) The title of the paper is changed as follows:   Home immunization with palivizumab. A randomized pilot study describing safety aspects and parents' preferences.   (II) The word "study" is replaced with "pilot study" throughout the paper.   For example: In Discussion (page7, line 305) change "The present study" to "The present pilot study";  In Strengths and Limitations (page 9, lines 362 and 366) change "The study" to "The pilot study" Make other additional changes from "study" to "pilot study" throughout the paper.   (iii) The conclusions use definitive language, e.g. "study shows".  I want to emphasize that the sample size used in this pilot study does not allow for definitive conclusions. I suggest changing Conclusions as follows: This pilot study indicates that home immunization with palivizumab may be feasible if consideration is paid to safety aspects and the risk of AE. The pilot study also indicates that parents may prefer immunization at home if they were allowed to choose.    (iv) Please study a section in the paper showing a comparison between the two groups of cost per patient.  If detailed cost information was not collected then, at a minimum, provide relative costs per patient for the two groups, i.e, on an average, whether treating a patient in one group was more expensive than that in the other group and by how much.    

Author Response

Thank you and the reviewers for the valuable comments on our manuscript. We are happy that you found the manuscript well written and to be of interest for readers. Please find below our respond point by point. As requested, the revisions made to the manuscript are marked up using the “Track Changes” function.

Reviewer’s comments to authors and our response

Reviewer 2 comments;

This paper reports a pilot study that was randomized. The total number of patients in the trial was 43, which is within the typical range for a pilot study.  On page 2, line 67, the authors mention that it is a pilot study, but that is the only location I found that the "pilot study" aspect of the study was mentioned. The title of the study gives an impression that it was a randomized controlled trial that provided definitive conclusions.  However, the small sample size of the pilot study does not provide data to derive statistically significant conclusions. I suggest the following revisions in the paper: (I) The title of the paper is changed as follows:   Home immunization with palivizumab. A randomized pilot study describing safety aspects and parents' preferences.   (II) The word "study" is replaced with "pilot study" throughout the paper.   For example: In Discussion (page7, line 305) change "The present study" to "The present pilot study";  In Strengths and Limitations (page 9, lines 362 and 366) change "The study" to "The pilot study" Make other additional changes from "study" to "pilot study" throughout the paper.     

Authors’ reply: To clarify that the study was a pilot study changes are done throughout the paper ( line 18, 64,297, 299,371, 373, 375, 377, 378, 381, 383 )including the title of the paper (line 2).

 (III) The conclusions use definitive language, e.g. "study shows".  I want to emphasize that the sample size used in this pilot study does not allow for definitive conclusions. I suggest changing Conclusions as follows: This pilot study indicates that home immunization with palivizumab may be feasible if consideration is paid to safety aspects and the risk of AE. The pilot study also indicates that parents may prefer immunization at home if they were allowed to choose. 

Authors’ reply: Conclusion of study result are changed as suggested (line 381-384).

 (IV) Please study a section in the paper showing a comparison between the two groups of cost per patient.  If detailed cost information was not collected then, at a minimum, provide relative costs per patient for the two groups, i.e, on an average, whether treating a patient in one group was more expensive than that in the other group and by how much.                                                                                                                                            Authors’ reply: We understand that calculation of cost are of great interest for the readers. Unfortunately an economic analyzes was not the purpose for this pilot study. Nevertheless, we collected data of cost for drugs and estimated the time spent for personnel and parents for help with calculation of our costs. A section about costs has been added in the discussion (line 346-353) including new references added to the list of references.                                    We also add information that the cost for the drug was covered by the clinic (line 99). 

Please let us know if this is enough clarifying.

Thank you very much for your valuable feedback!